# Development of Nano-Sized Copper-Deposited Antimicrobial Air Filters Using a Mixed Melt-Blown Process

**DOI:** 10.3390/nano13142071

**Published:** 2023-07-14

**Authors:** Kyung Hwan Lee, Jun Young Yoo, Chan Jung Park, Kang Ho Ahn

**Affiliations:** 1Department of Mechanical Convergence Engineering, Hanyang University, 222 Wangsimni-ro, Seongdong-gup, Seoul 04763, Republic of Korea; khahn@hanyang.ac.kr; 2Coway Environmental Technology Research Institute, Coway R&D Center, Seoul National University Research Park 1, Gwanak-ro, Gwanak-gu, Seoul 08826, Republic of Korea; cwolf98@coway.com (J.Y.Y.); 9056@coway.com (C.J.P.)

**Keywords:** antimicrobial, air filter, copper, masterbatch, melt-blown process

## Abstract

Air purification devices, such as air purifiers, provide fresh air by filtering out airborne pollutants, dust, and other harmful substances using various filter systems. While air filters are generally effective in filtering pollutants such as dust, they encounter a challenge when filtering harmful microorganisms such as mites, bacteria, mold, and viruses. These microorganisms, which are present in public transport and public indoor spaces, tend to proliferate on the surface of the filter media, eventually reintroducing themselves into the air or causing unpleasant odors. To address this issue, herein, copper particles were prepared as one masterbatch and deposited on polypropylene (PP) pellets through plasma vacuum deposition to effectively filter dust and microorganisms and prevent their growth on the surface of the filter media. After adding 3–10 wt.% of the masterbatch to conventional PP pellets to fabricate a filter media, the distribution of copper on the surface of the filter media was observed through a scanning electron microscope. To verify the safety and effectiveness of the antimicrobial material, the filter media containing antimicrobial particles was tested using *Escherichia coli* (*E. coli*) and *Staphylococcus aureus* through a filter emission test.

## 1. Introduction

People spend most of their time indoors, with individuals spending over 21 h a day inside. Thus, the indoor air quality can potentially affect people’s health, particularly their respiratory and vascular health [1]. According to the analysis performed by the Ministry of Land, Infrastructure, and Transport on commuting time, commuters spend ~3 h daily using public transportation, with over 56% of commuters in the Seoul metropolitan area relying on the subway as their primary mode of transportation. As urban railways witness an increase in usage, managing an optimal indoor air quality in railways has become crucial.

Recently, the health risks posed by indoor airborne bioaerosols have considerably increased. Respiratory illnesses caused by bacteria, fungi, or viruses, such as severe acute respiratory syndrome (SARS) and COVID-19, are spreading, with many infections occurring at indoor spaces including on public transportation and at preschools, offices, and gyms.

Currently, public places (subways, offices, etc.) are incorporating indoor air purification systems by installing high-efficiency particulate air filters within their heating, ventilation, and air conditioning systems. These filters effectively eliminate airborne bioaerosols [2]. More recently, air purification devices have been installed in each subway cabin to remove fine dust and bioaerosols, thereby improving the indoor air quality in subway cabins. Most of these air purifiers use filter systems, which, if not properly maintained or replaced, can accumulate dust and create an environment conducive to microbial growth. Furthermore, the improper operation of contaminated filters can cause secondary contamination [3]. To solve this problem, studies have explored the incorporation of antimicrobial properties into filters. It is essential to find a method that provides antimicrobial protection without compromising the performance of the filter.

Herein, a method to maximize the antimicrobial performance through the even distribution of copper particles throughout the filter media was sought using copper to impart an antimicrobial function to a filter. According to the US Environmental Protection Agency, copper is a safe antiviral metal ion [4], and several studies have demonstrated the antiviral effects of copper [5]. Recent studies have observed the viral inactivation of aerosolized SARS-coronavirus 2 and SARS-coronavirus 1 viruses within 4–8 h of exposure to copper surfaces [6]. The antiviral properties of copper can be attributed to its ability to damage viral cell membranes and nucleic acids, leading to viral inactivation [7,8].

To achieve the desired antimicrobial function without compromising the inherent performance of the filter, a method of depositing copper onto a dust filter media was adopted. Copper particles, which are known for their proven safety and excellent antimicrobial performance, were evenly coated on the surface of polypropylene (PP) pellets, which are used in making dust filter media, using electric plasma. These copper-deposited pellets were then transformed into a masterbatch, and the filter media was fabricated by mixing and spinning them with the regular PP pellets without copper. A dust filter generally comprises dust filter media and a support layer. Herein, the copper antimicrobial treatment was applied to the material to make the dust filter media, and the distribution and content of copper particles on the surface of the filter media were analyzed to ensure that the copper particles were evenly distributed on the filter media. After the even distribution of particles was confirmed, the antimicrobial and antifungal performance of the filter was analyzed, and the efficiency and differential pressure performance of the antimicrobial filter were compared with that of an existing original filter.

## 2. Materials and Methods

### 2.1. Method for Applying Antimicrobials and Test Antimicrobial Particle Dispersion

A test was performed to deposit and coat copper particles as an antimicrobial agent on a PP pallet to produce dust filter media. As shown in Figure 1, under the pressure of 10^−5^–10^−6^ Pa in a vacuum chamber, a copper plate of 20 cm × 10 cm × 1.5 cm was fixed to the upper part of the chamber. Copper particles were then deposited onto the PP pellets using a 65 kW high-voltage plasma. The resulting copper-deposited PP pellets were used as a masterbatch to fabricate antimicrobial dust filters.

To fabricate an antimicrobial dust filter media, a melt-blown process using the antimicrobial masterbatch and conventional PP pellets was employed. The melt-blown process (Figure 2) is a one-step technique that converts polymer resin into a nonwoven web or tow comprising low-diameter fiber. After spinning, an antimicrobial filter media was produced by mixing the polymer resin with the masterbatch.

To analyze the uniformity of copper particle coating on the antimicrobial dust filter media, one dust filter medium was randomly selected for each mixing ratio (masterbatch:PP) and its surface was analyzed via scanning electron microscopy (SEM). Based on the SEM analysis, the degree of even distribution of antimicrobial particles and the content of copper particles on the media surface was determined.

### 2.2. Performance Evaluation Test of Antimicrobial Filter

#### 2.2.1. Antimicrobial Performance via Antimicrobial Treatment Concentration

Following the guidelines of KS K 0693:2016, the antimicrobial performance test was performed to evaluate the efficacy of the antimicrobial dust filter media with various concentrations of copper distribution. A comparison was made between the antimicrobial masterbatch-enhanced filters and conventional filters. *Escherichia coli* (*Escherichia coli* ATCC 25922) was used as the main test strain to assess the antimicrobial performance across various concentration ranges, and the antimicrobial performance was analyzed for various treatment concentrations [9,10].

#### 2.2.2. Antimicrobial and Antifungal Testing of the Antimicrobial Filter

To further verify the antimicrobial effectiveness of the antimicrobial dust filter fabric, additional antimicrobial tests were performed with *Staphylococcus aureus* (ATCC 6538) and *Klebsiella pneumoniae* (ATCC 4352) strains as target strains. The antimicrobial test was performed by inoculating the test strain onto 4 cm × 4 cm specimens of the antimicrobial filter media and then incubating them. Subsequently, the antimicrobial performance was assessed by monitoring the antimicrobial rate for 18 h after applying the test strain [11].

Additionally, an antifungal test was performed using three samples—an untreated dust filter, a copper antimicrobial-treated filter, and a commonly used zirconium antimicrobial-treated filter. *Aspergillus niger* strains were spread on a *sabouraud dextrose agar* medium, and each of the three prepared samples was placed at the center of the medium. Furthermore, the fungal resistance of the three samples was observed for two weeks [12,13].

#### 2.2.3. Airborne Bacteria Removal Performance of the Antimicrobial Filter

To verify the effectiveness of the antimicrobial filter in removing airborne bioaerosols in an air purification system, we measured the removal performance over time after suspending bacteria in a 30 m^3^ chamber, as shown in Figure 3. An air purifier with the antimicrobial filter was placed at the center of the chamber under the test environment at 23 °C and a relative humidity of 50%. To simulate real-life conditions, the test strain *Staphylococcus aureus* (*Staphylococcus* ATCC 6538) was sprayed inside. Subsequently, the initial airborne bacteria concentration immediately after spraying the test strain was compared with the airborne bacteria concentration after the air purifier was run. As a control test, the test strain was sprayed into the chamber without the air purifier, and the concentration of airborne bacteria was checked at the beginning and after the same duration as the runtime of the air purifier to test the removal performance of airborne bacteria in the chamber space.

### 2.3. Basic Performance of the Filter and Antimicrobial Safety Test

To assess the effect of the antimicrobial agent on the performance of the dust filter, the fundamental characteristics of the dust filter, including the filtering efficiency and the differential pressure, were compared with those of the filter without the antimicrobial agent. To evaluate the performance of the filter media, the filtering efficiency for filtering 0.3 μm particles and the pressure of the flowing air in the media were measured.

Regarding the safety of the antimicrobial dust filter, an emission test was performed on a filter coated with the antimicrobial agent in accordance with the regulations for testing household chemical products subject to safety checks, as outlined in the National Institute of Environmental Research’s notification No. 2021-12. A 30 cm × 30 cm filter was installed in the wind tunnel (Figure 4) and continuously operated for 96 h with an operating flow rate of 10 m^3^/min. The emission quantity of the antimicrobial agents was determined by analyzing the content of the antimicrobial agents in a conventional filter and the tested filter, and the reduction in the content of antimicrobial agents was obtained. The antimicrobial content was measured before and after the filter test using a mass spectrometer.

## 3. Results

### 3.1. Antimicrobial Dispersion Test Results

Figure 5a shows a sample photograph of the copper-deposited PP pellets using high-voltage plasma in a vacuum chamber. Figure 5b shows a photograph of the masterbatch of copper-deposited PP pellets. A filter medium for dust collection was fabricated via a melt-blown process using the aforementioned masterbatch.

The 1000-fold magnified SEM image depicted in Figure 6 reveals the presence of copper particles coating the filter medium, achieved using the masterbatch. Copper particles with a size of 20–200 nm were coated on the fiber surface.

Figure 7 displays the SEM analysis of the surface of the filter without copper antimicrobials and with varying percentages (3%, 6%, and 10%) of the copper-deposited masterbatch. The images confirm the even distribution of copper particles on the surface of the media specimens. As the mixing ratio increased, the content of copper particles also increased, resulting in concentrations of 1.2, 2.2, and 4.3 wt.% for the 3%, 6%, and 10% weight ratios of the copper-deposited masterbatch, respectively.

### 3.2. Antimicrobial Performance

#### 3.2.1. Antimicrobial Effectiveness Based on the Concentration of Antimicrobial Agents

After confirming the distribution of antimicrobial copper particles in the dust filter media, the antimicrobial effect was evaluated based on the treatment concentration of the antimicrobial agent. As shown in Figure 8 and Table 1, the masterbatch content of 3% with the copper-deposited antimicrobial agent exhibited an antimicrobial effect of 98.3% in the antimicrobial test. An excellent antimicrobial effect of 99.9% was confirmed in the melt-blown medium that used 6% of the copper-deposited masterbatch, while an excellent bacterial reduction rate of >99.9% was confirmed in the melt-blown medium that used 10% of the copper-deposited masterbatch. Overall, a good antimicrobial performance was observed in the melt-blown medium that used >6% of the copper-deposited antimicrobial masterbatch.

#### 3.2.2. Optimized Concentration Filtration Antibacterial and Antifungal Test Results

Figure 9 illustrates an experiment performed to further verify the antimicrobial effect of the melt-blown filter medium that used a 6% antimicrobial masterbatch. The test standard used in this study was AATCC100, a quantitative testing methodology used to determine the efficacy of antibacterial finishes applied to textile materials. The experimental results revealed excellent bacterial reduction rates of 99.9% against *Staphylococcus aureus* ATCC 6538 and *Klebsiella pneumoniae* ATCC 4352 (Figure 9). These results confirmed the uniform distribution of copper antibacterial particles within the media and excellent antibacterial effects (Table 2).

Figure 10 shows the antifungal performance evaluation of a sample of melt-blown media using a 6% copper-deposited masterbatch. To compare the antifungal performance of copper, a melt-blown test was also performed using a zirconium treatment. The image shows that halo zones were formed in the antibacterial melt-blown sample. These halo zones corresponded to areas within the media where mold growth is hindered because of the presence of an agent that inhibits the growth.

In the control group, *Aspergillus niger* fungi extensively proliferated across the plate, while the specimen of the melt-blown media exhibited limited fungi growth, with a diameter of 19 mm, implying that the antifungal performance of the melt-blown media with copper is better than that of the commonly used zirconium antimicrobial material.

#### 3.2.3. Suspended Bacteria Test Results

To evaluate the effectiveness of an air purifier equipped with a melt-blown antimicrobial filter containing 6% of an antimicrobial masterbatch, *Staphylococcus aureus* (ATCC 6538) was suspended in a 30 m^3^ chamber. The concentration of suspended bacteria was measured before and after a 1 h operation of the purifier.

The results presented in Table 3 reveal a reduction rate of 99.9% in the concentration of suspended *Staphylococcus aureus* bacteria.

### 3.3. Basic Performance of the Filter and Safety Test Results

#### 3.3.1. Basic Performance (Differential Pressure and Efficiency)

To assess the effect of antimicrobial treatment on the performance of the dust filter, the differential pressure and filtering efficiency of the antimicrobial-treated filter were compared to those of the original filter without antimicrobial treatment, as shown in Table 4. The results indicated that the inherent performance of the filter was not affected by the microscopic size of the antimicrobial copper particles during the antimicrobial treatment.

#### 3.3.2. Safety Test Results

During the 96 h filtering test performed in the wind tunnel with a wind flow of 10 m^3^/min, the reduction in the amount of copper, which serves as an antibacterial agent for filters, was 17.4 mg in the filter media. This value is considerably lower than the allowable decrease of 70 mg, as specified by the safety verification standard for household chemical products. This can be attributed to the considerable stability of the copper antimicrobial agent in the filter media, indicating that the antimicrobial agent does not reduce easily.

## 4. Conclusions

In this study, a high-voltage plasma deposition method was used to stably and evenly deposit an antibacterial agent onto PP pellets, which are commonly used for manufacturing dust filter media. The PP pellets deposited with the antibacterial agent were transformed into a masterbatch, and were then mixed-spun with existing pellets to produce the filter media. The surface of the resulting filter media was analyzed using SEM, revealing an even distribution of antibacterial copper particles. Additionally, copper contents of 1.2, 2.2, and 4.3 wt,% were observed depending on the concentration of the antibacterial mixed masterbatch during the filter media fabrication. The antibacterial performance of the melt-blown media was evaluated for various concentrations. It was observed to be high for a mixture concentration of 6% and above. These filter media exhibited exceptional antibacterial effects against *S. aureus*, *E. coli*, and *K. pneumoniae* at a 6% mixing level. Regarding antifungal performance, the size of the inhibition zone revealed that the antifungal performance against *Aspergillus niger* was more than five times higher than that of conventional antimicrobials. The antimicrobial-treated filter media were incorporated into a dust filter, and the *S. aureus* removal rate in the test chamber with the filter was as high as 99.9% within 1 h. When using antimicrobial-treated filters, the filtering efficiency and differential pressure performance of the filter were compared to those of an original filter without antimicrobial treatment, and the performance was observed to be equivalent or better. This reveals that the antimicrobial treatment had no adverse impact on the performance of the original filter. In the chemical safety test, the amount of antimicrobial agent lost in the filter was 17.4 mg, which was considerably lower than the permissible standard of 70 mg. Considering the possibility of the secondary contamination of dust filters, the use of antibacterial dust filters with an even distribution of copper antibacterial agents, that have been confirmed to be safe in public places or subway stations, is recommended.

## Figures and Tables

**Figure 1 nanomaterials-13-02071-f001:**
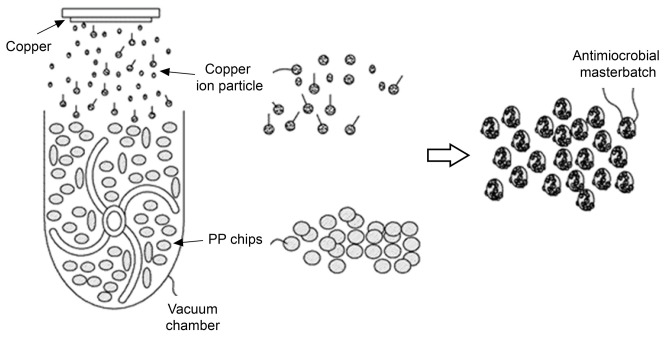
Experimental setup for the deposition of copper particles on polypropylene PP pellets.

**Figure 2 nanomaterials-13-02071-f002:**
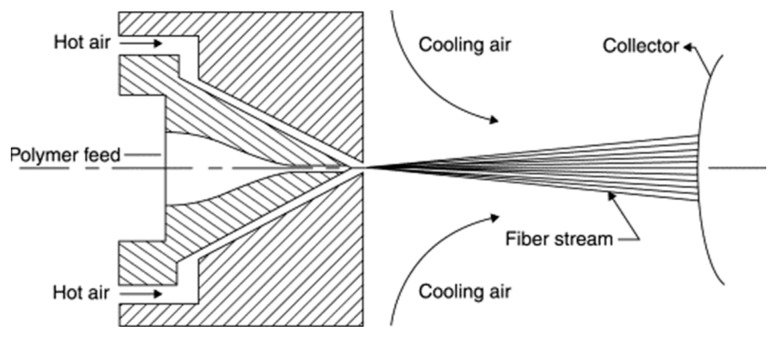
Schematic of the melt-blown process.

**Figure 3 nanomaterials-13-02071-f003:**
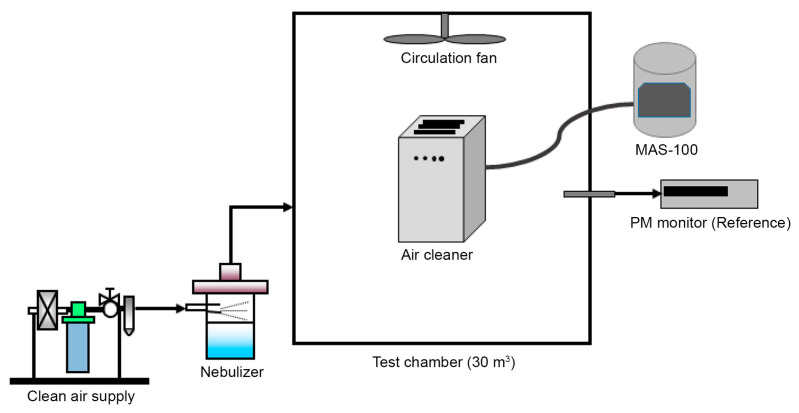
Experimental setup for airborne bacteria removal performance.

**Figure 4 nanomaterials-13-02071-f004:**
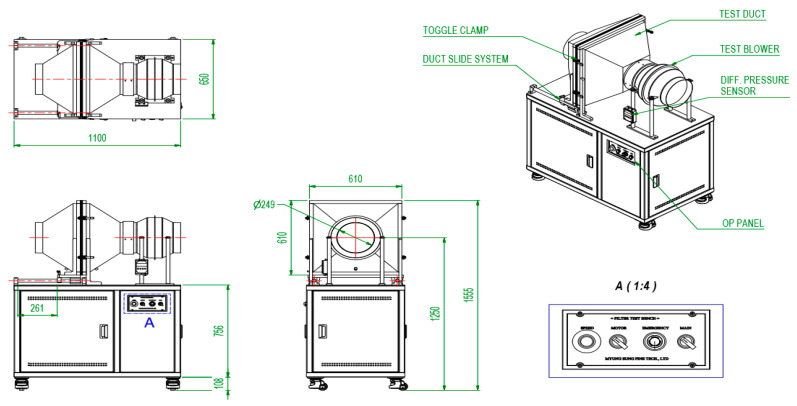
Experimental setup of the emission measurement station.

**Figure 5 nanomaterials-13-02071-f005:**
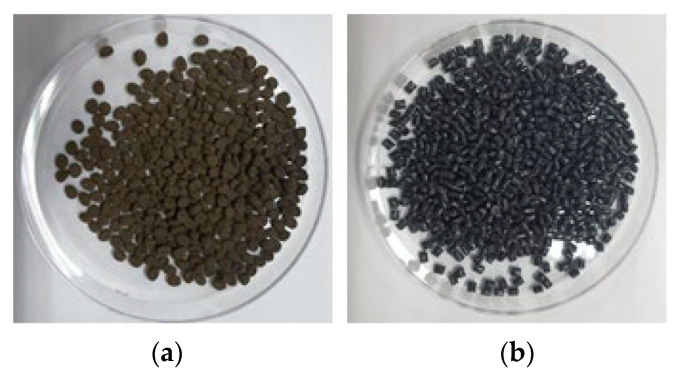
Photographs of (**a**) the copper-deposited PP pellets and (**b**) the masterbatch.

**Figure 6 nanomaterials-13-02071-f006:**
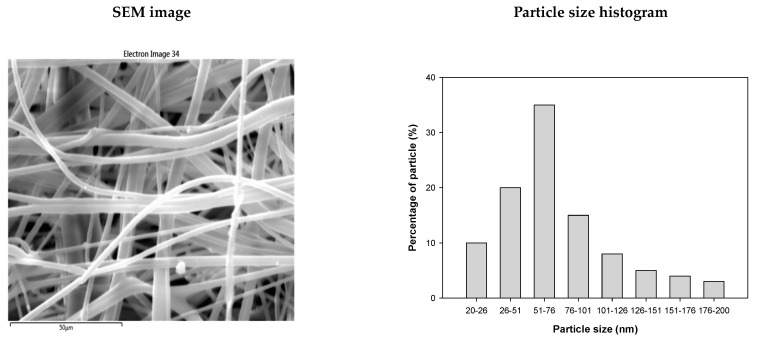
Scanning electron microscopy (SEM) image and particle size distribution of melt-blown fiber containing copper particles.

**Figure 7 nanomaterials-13-02071-f007:**
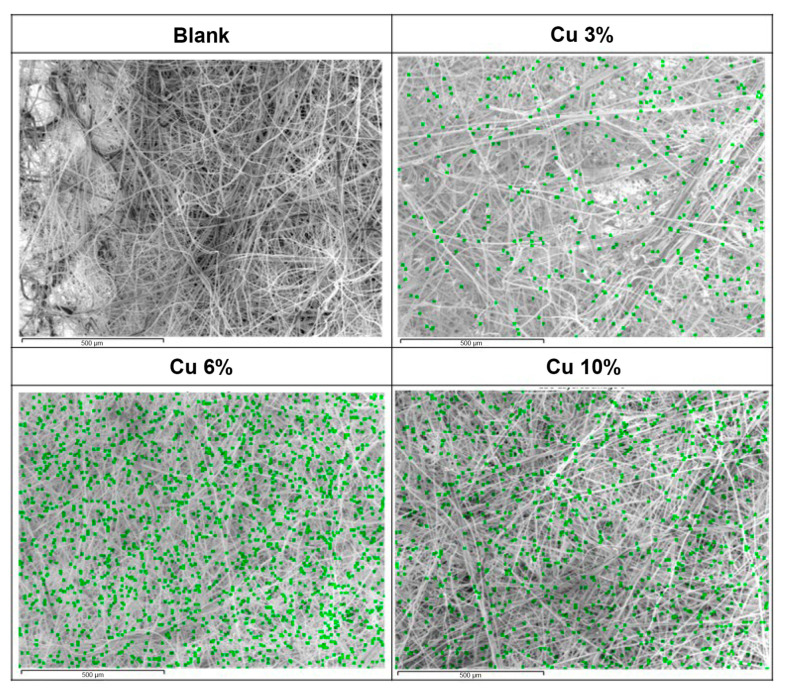
Surface analysis of antibacterial media for different concentrations using SEM.

**Figure 8 nanomaterials-13-02071-f008:**
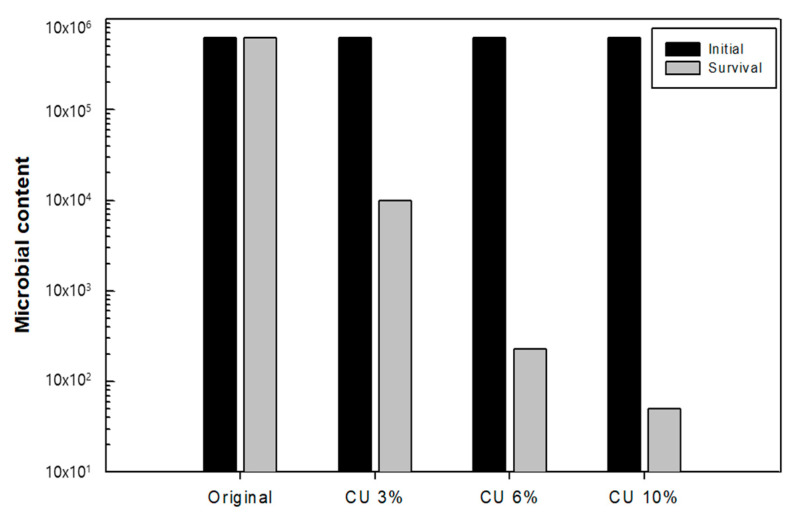
Antibacterial performance with different concentrations of the antimicrobial agent.

**Figure 9 nanomaterials-13-02071-f009:**
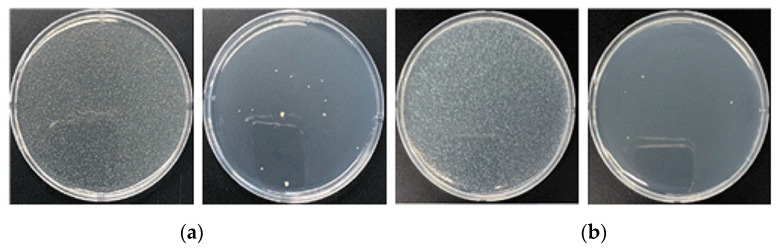
Antimicrobial performance with 6% antimicrobial agent mixed media. (**a**) *Staphylococcus aureus* and (**b**) *Klebsiella pneumoniae*.

**Figure 10 nanomaterials-13-02071-f010:**
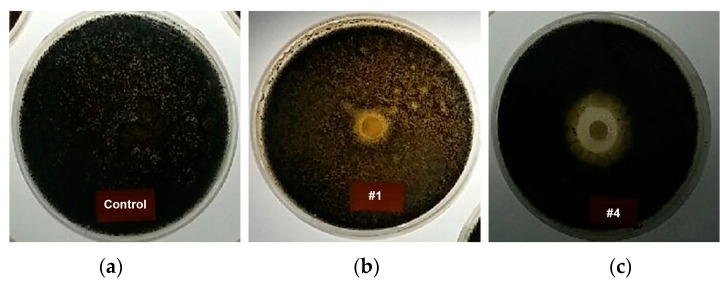
Photographs of antifungal activities using 6% antimicrobial agent mixed media. (**a**) Original, (**b**) zirconium, and (**c**) copper.

**Table 1 nanomaterials-13-02071-t001:** Antimicrobial performance at different concentrations of the antimicrobial agent.

	Blank	Original	Cu 3%	Cu 6%	Cu 10%
Concentration (cfu/mL)	6.6 × 10^5^	6.7 × 10^5^	8.8 × 10^3^	1.2 × 10^2^	45
Inhibition rate (%)	-	0	98.7	99.9	99.9

**Table 2 nanomaterials-13-02071-t002:** Antimicrobial reduction and removal rate with 6% antimicrobial agent mixed media.

Test Item	Test Result	
	Control	Sample
*Staphylococcus*	6.6 × 10^5^	2.8 × 10^2^ 99.9%
*Klebsiella pneumoniae*	3.0 × 10^6^	<10 99.9%

**Table 3 nanomaterials-13-02071-t003:** Airborne bacteria removal performance of the melt-blown antimicrobial filter containing 6% of an antimicrobial masterbatch.

Test Item	Test Result			Test Condition
	Before Operation (CFU/m^3^)	After Operation (CFU/m^3^)	Removal Rate (%)	
*S. Aureus*	1.2 × 10^4^	<10	99.9	23.0 °C ± 0.2 °C 50.5% ± 3.0% R.H.

**Table 4 nanomaterials-13-02071-t004:** Results of media’s basic performance.

Test Item	Test Result		Test Method
	Efficiency (%)	Resistance (Pa)	
H13 (Original)	99.98	33.7	EN 1822-3 (Flat sheet media)
H13 (Cu 3%)	99.98	34.2	
H13 (Cu 6%)	99.97	32.2	
H13 (Cu 10%)	99.97	30.6	

## Data Availability

The data that support the findings of this study are available on reasonable request from the corresponding author.

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
