# Peer review of "Development of Nano-Sized Copper-Deposited Antimicrobial Air Filters Using a Mixed Melt-Blown Process"

_nanomaterials, 2023, doi:10.3390/nano13142071_

Round 1

Reviewer 1 Report

This manuscript must be  improved before possible openly publication in future, and some advice are presented here for authors' further considrations: (i) please provide the higher-solution SEM images for clearly visualizing cu particles both on the surface of PP pellets and melt-brown filters. And please give description on how many sizes of the cu particles and how about the status of  Cu paticles when distributed whithin PP filters? (ii) Reorganize and combine the related antibacterial performances in 3.2.1 and 3.2.2 parts of this present manuscript.

Author Response

[Reviewer 1]

This manuscript must be  improved before possible openly publication in future, and some advice are presented here for authors' further considrations:

à We appreciate your suggestion.

(i) please provide the higher-solution SEM images for clearly visualizing cu particles both on the surface of PP pellets and melt-brown filters. And please give description on how many sizes of the cu particles and how about the status of  Cu paticles when distributed whithin PP filters?

à We included Figure 4, which shows a photograph of fiber coated with copper particles. The size of the copper particle size was determined using an SEM micrograph, which revealed sizes ranging from 0.1 to 10 µm. We have included pictures and descriptions within the text.

(ii) Reorganize and combine the related antibacterial performances in 3.2.1 and 3.2.2 parts of this present manuscript.

à Findings from Sections 3.2.1 and 3.2.2 show different results. In Section 3.2.1, the experiment focused on optimizing the concentration of antimicrobial treatment. However, Section 3.2.2 presents the result of testing the antibacterial and antifungal performance of the filter, which was optimized for the antimicrobial treatment concentration of 6%. The title of Section 3.2.1 has been revised for classification.

Original title: 3.2.1 Antimicrobial test results by antimicrobial agent treatment concentration

Revised title: 3.2.1 Antibacterial test results by concentration for antimicrobial optimization

Reviewer 2 Report

This is a very interesting and potentially important paper to develop antimicrobial polymers for filter applications.  The concept is sound, and importantly uses copper (rather than silver) for its effect - this is good for reasons of toxicity and cost.

However, the manuscript could be improved by:

* line 81 - please give more details of filter preparation by melt blown process

* line 118 - would a better control be to use a standard filter in the air purifier, rather than having no purfier at all?

* line 129 - please describe experimental set-up and give data

* line 136 - please describe experimental set-up and give data

* Figure 5 - vertical scale "Microbial content" should read "Cell count"?

* Table 1 - title should read "Cell count and % reduction for modified materials" ?

* Figure 7 - not sure what I am supposed to look at here?

* line 188 - this data shows removal of S aureus, but without a control of untreated filter medium, it does not demonstrate efficacy of the Cu treatment

* line 205 - I would like to see more of the primary data from this experiment.  What is the size of filter material which was examined? How was the Cu analysis done?

Author Response

[Reviewer 2]

This is a very interesting and potentially important paper to develop antimicrobial polymers for filter applications.  The concept is sound, and importantly uses copper (rather than silver) for its effect - this is good for reasons of toxicity and cost

àWe appreciate your invaluable advice.

However, the manuscript could be improved by:

* line 81 - please give more details of filter preparation by melt blown process

à A photograph of the melt-blown process has been added in the revised manuscript. In addition, detailed explanations regarding production using the melt-blown process have been included.

* line 118 - would a better control be to use a standard filter in the air purifier, rather than having no purfier at all?

à In the case of removing airborne bacteria from the chamber, the experiment uses an air purifier instead of just testing a single filter.

* line 129 - please describe experimental set-up and give data

à A picture depicting the experiment has been included, along with a paraphrase in the text. (Figure 4)

* line 136 - please describe experimental set-up and give data

à An image of the experimental result was added and included the details of the result in Section 3.1.

* Figure 5 - vertical scale "Microbial content" should read "Cell count"?

à The table now includes a quantitative expression of the bacteria and corresponding removal rates.

* Table 1 - title should read "Cell count and % reduction for modified materials" ?

à The title of Table 1 has been modified.

* Figure 7 - not sure what I am supposed to look at here?

à The purpose and test method for Figure 7 have been added in the text, along with a description of the antifungal test.

* line 188 - this data shows removal of S aureus, but without a control of untreated filter medium, it does not demonstrate efficacy of the Cu treatment

à We agree with your statement. However, the purpose of this experiment was to confirm the effectiveness of the antibacterial filter in removing bacteria from the air. We hope this explanation resolves your concern.

* line 205 - I would like to see more of the primary data from this experiment.  What is the size of filter material which was examined? How was the Cu analysis done?

à The test method has been added to the main text, and this test method involves quantitatively measuring the consumption of antibacterial agent by antibacterial filter to estimate the emission amount. This is method involves the analysis of the antibacterial content in the filter before its usage, employing a mass spectrometer.

Reviewer 3 Report

The author proposed an interesting research hypothesis, but the author did not provide enough data to confirm the author's conjecture. It is suggested that the author add more data to explain. The pictures in this paper are rough, as shown in Figure 4, Figure 5, Figure 6 and Figure 7. At the same time, the design of some experiments is unreasonable, the necessary control experiments are lacking, as shown in Figure 6 and Figure 7, and the experimental data lacks statistical analysis, as shown in Figure 5.

Moderate editing of English language required

Author Response

[Reviewer 3]

The author proposed an interesting research hypothesis, but the author did not provide enough data to confirm the author's conjecture. It is suggested that the author add more data to explain.

The pictures in this paper are rough, as shown in Figure 4, Figure 5, Figure 6 and Figure 7. At the same time, the design of some experiments is unreasonable, the necessary control experiments are lacking, as shown in Figure 6 and Figure 7, and the experimental data lacks statistical analysis, as shown in Figure 5.

The number in the image has been modified, which could cause confusion. I would appreciate it if you could check the corresponding figure caption.

[Figure 4] Surface analysis of antibacterial media by concentration

àTo supplement the contents in Figure 4, an image of copper particles coated on fiber was added. The copper particle size was confirmed to be ranging from 0.1 to 10 µm using the SEM micrographs. Corresponding descriptions are added to the text.

[figure 5] Results of antibacterial performance by concentration

à The table was updated with quantitative data on the number of bacteria and removal rates. The antibacterial activity of the control and the experimental groups was compared and analyzed.

[figure 6] Antimicrobial performance by anti agent 6% mixed media. A. Staphylococcus, B. Klebsiella pneumoniae

à The antibacterial test method for fibers and fabrics was specified, and supplementary explanations were added for the test standards.

[figure 7] Picture of antifungi activities by anti agent 6% mixed media. A. Origina, B. Zirconium,

à The purpose and test method for Figure 7 have been added to the text. Furthermore, the description of antifungal test has been added in the revised manuscript.

Thank you for providing valuable feedback to improve our manuscript. I greatly appreciate your comments, and we have implemented several revisions based on your suggestions to enhance the overall quality of the study.

Round 2

Reviewer 1 Report

The quality of this present manuscript had been improved, but, I consider, some correction and modification are still not enough. In figure 6 added in this modified manuscript, the author dipicted the sizes of Cu deposited onto PP fibres is in 1~10 micrometer range, but not in nanometer range. this is not consist with the nanosize Cu as said in Title of manuscript. In addition, the author also is not too successful in reorganize the part in anitibacterial performance .

Author Response

[Reviewer 1]

The quality of this present manuscript had been improved, but, I consider, some correction and modification are still not enough. In figure 6 added in this modified manuscript, the author dipicted the sizes of Cu deposited onto PP fibres is in 1~10 micrometer range, but not in nanometer range. this is not consist with the nanosize Cu as said in Title of manuscript. In addition, the author also is not too successful in reorganize the part in anitibacterial performance .

I acknowledge that in the previous version of the manuscript, particle distribution was was not accurately analyzed. However, I have reevaluated the particle distribution and included the updated findings in the manuscript. My sincere apologies for the oversight. Furthermore, I have provided the specific test results regarding antibacterial activity based on concentration. In particular, the manuscript now includes the comprehensive results of the antibacterial test performed at a concentration of 6%. These results provide valuable insights into the effectiveness of the antimicrobial treatment and its impact on bacterial reduction. Thank you for your understanding, and I apologize for any confusion caused in the previous version of the manuscript.

Reviewer 2 Report

The authors corrections have addressed my queries adequately

Modest correction is needed

Author Response

Thank you very much for your comment. This time, the paper as a whole has been reviewed and revised once again. I appreciate it.

Reviewer 3 Report

The authors did not respond to the comments favorably specially the lack of full characterization of the addressed Cu nanomaterials which is a key investigation On which bases did the author select the antiagent  6% media mix performed in in vitro studies to ensure the authenticity of the study.

Therefore, unfortunately I can not accept the paper in its current form. However, the manuscript  is worth reconsidering after improvement.

A complete English language editing is required for the following manuscript 

Author Response

The authors did not respond to the comments favorably specially the lack of full characterization of the addressed Cu nanomaterials which is a key investigation On which bases did the author select the antiagent  6% media mix performed in in vitro studies to ensure the authenticity of the study.

Therefore, unfortunately I cannot accept the paper in its current form. However, the manuscript  is worth reconsidering after improvement.

A complete English language editing is required for the following manuscript 

We have made a comprehensive revision of the English text. Furthermore, we have reanalyzed the particle distribution and incorporated the updated findings into the manuscript. Additionally, we have provided specific details on the test results regarding antibacterial activity based on concentration. Notably, the manuscript now presents the results of the antibacterial test conducted at a concentration of 6%. Thank you for your understanding.

Round 3

Reviewer 3 Report

The authors provided good responses to the reviewers' comments.